# Microbiota Transplantation in Day-Old Broiler Chickens Ameliorates Necrotic Enteritis via Modulation of the Intestinal Microbiota and Host Immune Responses

**DOI:** 10.3390/pathogens11090972

**Published:** 2022-08-26

**Authors:** Sarah J. M. Zaytsoff, Tony Montina, Valerie F. Boras, Julie Brassard, Paul E. Moote, Richard R. E. Uwiera, G. Douglas Inglis

**Affiliations:** 1Lethbridge Research and Development Centre, Agriculture and Agri-Food Canada, Lethbridge, AB T1J 4B1, Canada; 2Department of Agricultural, Food and Nutritional Science, University of Alberta, Edmonton, AB T6G 2R3, Canada; 3Department of Chemistry and Biochemistry, University of Lethbridge, Lethbridge, AB T1K 3M4, Canada; 4Chinook Regional Hospital, Alberta Health Services, Lethbridge, AB T1J 1W5, Canada; 5Saint-Hyacinthe Research and Development Centre, Agriculture and Agri-Food Canada, Saint-Hyacinthe, QC J2S 8E3, Canada

**Keywords:** necrotic enteritis, *Clostridium perfringens*, microbiota transplantation, metabolomics, immune responses

## Abstract

A microbiota transplant (MT) originating from mature adult chicken ceca and propagated in bioreactors was administered to day-old broiler chicks to ascertain the degree to which, and how, the MT affects *Clostridium perfringens* (*Cp*)-incited necrotic enteritis (NE). Using a stress predisposition model of NE, birds administered the MT and challenged with *Cp* showed fewer necrotic lesions, and exhibited a substantially higher α- and β-diversity of bacteria in their jejunum and ceca. Birds challenged with *Cp* and not administered the MT showed decreased *Lactobacillus* and increased *Clostridium sensu strico* 1 in the jejunum. In ceca, *Megamonas*, a genus containing butyrate-producing bacteria, was only present in birds administered the MT, and densities of this genus were increased in birds challenged with *Cp*. Metabolite profiles in cecal digesta were altered in birds administered the MT and challenged with the pathogen; 59 metabolites were differentially abundant following MT treatment, and the relative levels of short chain fatty acids, butyrate, valerate, and propionate, were decreased in birds with NE. Birds administered the MT and challenged with *Cp* showed evidence of enhanced restoration of intestinal barrier functions, including elevated mRNA of *MUC2B*, *MUC13*, and *TJP1*. Likewise, birds administered the MT exhibited higher mRNA of *IL2*, *IL17A*, and *IL22* at 2-days post-inoculation with *Cp*, indicating that these birds were better immunologically equipped to respond to pathogen challenge. Collectively, study findings demonstrated that administering a MT containing a diverse mixture of microorganisms to day-old birds ameliorated NE in broilers by increasing bacterial diversity and promoting positive immune responses.

## 1. Introduction

Microbiota transplantation has emerged as a therapeutic approach to treat a number of enteric diseases in human beings, and it is beginning to garner interest in agriculture. The process of microbiota transplantation typically involves the transfer of a complete microbial community that includes viruses, archaea, bacteria, fungi, protozoa, host cells, metabolites, and other debris [1]. Beneficial outcomes of microbiota transplantation can include restructuring microbial communities and increasing community diversity [2]. Furthermore, the introduction of novel microorganisms may stimulate the immune system in a manner that limits infection [2]. Thus, microbiota transplantation imparts colonization resistance by promoting positive microorganism-microorganism and microorganism-host interactions that displace pathogens and reduce pathologic outcomes [3].

Necrotic enteritis (NE) is primarily a disease of the small intestine of poultry incited by the bacterium, *Clostridium perfringens* that results in significant production losses [4]. The administration of in-feed antibiotics has previously controlled NE, but reductions in antibiotic use have resulted in an increased incidence of NE in broiler chicken production [5]. One antimicrobial alternative for NE that has been widely explored is probiotics. The administration of probiotics was shown to limit NE via competitive exclusion, stimulation of immune functions, and production of bacteriocin [6,7,8,9,10]. However, probiotics typically need to be administered continuously to be effective, which represents an obstacle for their use in production settings. The administration of a microbiota transplant (MT) to chicks successfully excluded *Salmonella enterica* subsp. *enterica* serovar Infantis from ceca [11,12]. Microbiota transplantation has also been considered in poultry in the context of enhancing production efficiency and determining if behavioral phenotypes can be passed through the microbiota [13,14,15]. However, the exploration of microbiota transplantation in poultry has yet to be considered as an alternative to antimicrobials for mitigating NE, a disease that primarily affects the proximal small intestine of chickens. Moreover, the mechanisms of colonization resistance conferred by microbiota transplantation in poultry are currently enigmatic. Given the demonstrated benefits that administering a limited number of bacterial taxa can have in limiting NE (i.e., probiotics), the application of a MT as a method to mitigate NE warrants investigation.

The types of sanitation strategies applied in different jurisdictions vary substantially. The Chicken Farmers of Canada specify that a “wet cleaning” take place at least once a year (i.e., involving removal of litter followed by the application of a liquid sanitizer), and that between flocks, dry cleaning by removal and replacement of litter be applied at minimum (with at least 14 days between flocks as a best practice) [16]. Therefore, in Canada and elsewhere, broiler chicks are often exposed to hygienic environments after placement in barns, particularly in situations where new litter has been introduced with or without liquid sanitation of barn surfaces, and this may result in a delayed exposure to microorganisms. Chickens populate their gastro-intestinal tract via ingestion of microorganisms in the environment (e.g., litter), and a diverse microbiota is established by 7 days-of-age [17], although a succession of bacterial taxa occurs over the course of the broiler production cycle [18]. It is possible that the external application of microorganisms to broiler chicks early in life may expedite the establishment of a stable enteric bacterial community structure, and thereby confer positive health benefits to the birds. Probiotics in poultry have predominantly focused on the use of species that are primary colonizers of the small intestine, which is rational as the impacts of NE are primarily manifested in the duodenum and jejunum. However, ceca are also important in the health of chickens as the cecal microbiota aids in digestion and promotes immune development. Furthermore, commonly used probiotics, such as *Lactobacillus* spp., can facilitate cross-feeding to microorganisms that utilize lactate to produce butyrate [19] within ceca. We hypothesized that the introduction of a cecal community of bacteria obtained from healthy donors to neonatal broiler chicks will promote beneficial alterations to bacterial community structure, and modulate metabolites in the intestine that aid in disease resistance. Moreover, we hypothesized that the administration of a MT to day-old chicks will promote immune development in birds resulting in enhanced resistance to NE later in life. To test these hypotheses, primary objectives of the study were to: (i) propagate cecal bacteria obtained from mature healthy broiler males within bioreactors; (ii) orally administer the MT generated within bioreactors to day-old birds; (iii) determine the impacts of the MT on colonization of the intestine by *C. perfringens*, and on the composition and function of enteric bacterial communities; and (iv) ascertain the influence of the MT on NE, including on host immune responses. To facilitate the evaluation of the impacts of treatments on acute disease, corticosterone was provided in the diet to mediate a stress response in birds and promote the onset of clinical NE upon challenge with *C. perfringens* [20].

## 2. Results

### 2.1. Propagation of Cecal Bacteria from Healthy Adult Donors in Bioreactors Affected the Community Structure

Propagation of the cecal microbiota with bioreactors temporally affected both the composition and structure of bacterial communities. In this regard, the α-diversity of bacteria in the bioreactor medium was reduced relative to the diversity of bacteria present in the cecal digesta of donor birds; Shannon’s α-diversity of bacterial communities within bioreactors ranged from 4.2 at day 1 to 3.7 at day 10, as compared to an α-diversity of 6.7 within cecal digesta. The β-diversity of bacterial communities also changed over time (*p* < 0.050) (Appendix A). Within cecal digesta, amplicon sequence variants (ASVs) representing 128 genera were observed, whereas the bioreactor matrix contained 52 genera. Bacteria within the genera, *Bacteroides*, *Rikenellaceae*, *Phascolarctobacterium*, and *Prevotellaceae* were less abundant in the bioreactors than in cecal digesta of donor birds (Appendix A). In contrast, bacteria within the genus, *Megamonas* were substantially more abundant in bioreactors relative to cecal digesta. 

### 2.2. Administration of the Microbiota Transplant Reduced Pathologies

Experimental treatments were: (i) birds not administered the MT and not challenged with *C. perfringens* (i.e., Control treatment, n = 6); (ii) birds administered the MT and not challenged with *C. perfringens* (i.e., MT treatment; n = 6); (iii) birds not administered the MT and challenged with *C. perfringens* (i.e., NE treatment; n = 6); and (iv) birds administered the MT and challenged with *C. perfringens* (i.e., MT + NE treatment; n = 6). Treatment iii and iv birds were administered corticosterone in their diet. Only birds administered corticosterone and challenged with *C. perfringens* exhibited necrotic lesions (i.e., NE and MT + NE treatments) (Figure 1A,B). However, birds challenged with *C. perfringens* but not administered the MT (i.e., NE treatment) exhibited higher (*p* ≤ 0.006) necrotic lesion scores in comparison to Control and MT treatment birds (Figure 1B). Birds challenged with *C. perfringens* and administered the MT (i.e., MT + NE treatment) exhibited lower (*p* = 0.037) lesion scores in comparison to NE treatment birds. Densities of *C. perfringens* as determined by qPCR were highest (*p* ≤ 0.044) in jejunal digesta of NE treatment birds (Figure 1C), and rod-shaped bacteria characteristic of the pathogen were observed in association with the epithelium of birds challenged with the pathogen and not administered the MT; the bacterial cells were pleomorphic bacilli with rounded ends (≈3–8 µm × 0.4–1.2 µm in size) (Figure 1D). Although a treatment effect was not observed (*p* ≥ 0.179), cumulative total histopathologic change scores for MT + NE treatment birds trended lower than for NE treatment birds (Figure 1E). This was primarily due to lower degrees of villar atrophy, bacterial invasion, and mucosal necrosis in the jejunum of NE treatment birds relative to other treatments (Figure 1E and Appendix A).

### 2.3. The Enteric Bacterial Community Structure was Altered in Birds Administered the Microbiota Transplant ± Clostridium perfringens Challenge

The administration of the MT resulted in larger numbers of ASVs in the jejunum (78 and 53 ASVs with and without MT administration, respectively) and ceca (197 and 123 ASVs with and without MT administration, respectively). Despite the decrease in *Bacteroides and Phascolarctobacterium* within the bioreactors, these taxa were observed to predominate in the ceca of birds administered the MT (Appendix A). *Megamonas*, a bacterial genus that was enriched within the bioreactors relative to the cecal digesta inoculum, was also a prominent colonizer of the ceca of birds administered the MT. Overall, α-diversity was higher in the jejunal digesta (*p* < 0.001) and cecal digesta (*p* < 0.001) of birds administered the MT (Figure 2A,B). Birds challenged with *C. perfringens* and administered the MT (i.e., MT + NE treatment) exhibited a decreased (*p* = 0.030) α-diversity within cecal digesta relative to birds administered the MT alone (i.e., MT treatment). Although the structure of bacterial communities in the jejunal digesta of birds administered the MT were similar, the community structure differed (*p* ≤ 0.026) among all treatments within the jejunum (Figure 2C). In ceca, bacterial community structures differed (*p* = 0.001) only between the ± MT administration treatments (Figure 2D). Challenge with *C. perfringens* altered (*p* = 0.043) bacterial community structures in ceca, but only for birds that were not administered the MT (i.e., NE treatment).

The observed change in α- and β-diversity was associated with a conspicuous change in the relative abundance of multiple bacterial taxa in the jejunum and ceca (Figure 3A,B). The relative abundance of ASVs identified as *Lactobacillus* were lower (*p* ≤ 0.036) in the jejunum of NE treatment birds in comparison to MT + NE treatment birds (Figure 3C). Conversely, the relative abundance of ASVs identified as *Clostridium sensu stricto* 1 was higher (*p* ≤ 0.008) in the jejunum of NE treatment birds relative to other treatment birds (Figure 3C); all the *Clostridium sensu stricto* 1 sequences were classified as *C. perfringens*, which is consistent with the qPCR data of pathogen abundance (Figure 1C). For cecal communities, the relative abundances of *Megamonas* and *Bacteroides* spp. were higher (*p* ≤ 0.031) in birds administered the MT (Figure 3D). Moreover, the relative abundance of *Megamonas* spp. was higher (*p* < 0.010) in the cecal digesta of MT + NE treatment birds relative to MT treatment birds.

### 2.4. The Metabolome of Cecal Digesta was Altered in Birds Challenged with Clostridium perfringens Following Administration of the Microbiota Transplant

Water-soluble metabolites were extracted from jejunal digesta, jejunal tissue, and cecal digesta, and subjected to ^1^H-Nuclear Magnetic Resonance (NMR) spectroscopy to identify changes in the metabolome in birds challenged with *C. perfringens* ± administration of the MT. In total, 432, 550, and 178 metabolite bins were created from the jejunal digesta, jejunal tissue, and cecal digesta spectra, respectively. Multivariate supervised orthogonal partial least squares discriminant analysis (OPLS-DA) score plots showed no significant changes in the metabolome due to the MT in either jejunal digesta (Q^2^ = 0.018, *p* = 0.274; R^2^ = 0.734, *p* = 0.121) or jejunal tissue (Q^2^ = −0.421, *p* = 0.685; R^2^ = 0.989, *p* = 0.032) as both models failed permutation testing. However, univariate testing identified that three metabolites in jejunal digesta and seven metabolites in jejunal tissue of birds administered the MT were altered (Appendix A). Concentrations of citrate, dimethylamine, and 4,5-dihydroorotic acid were increased in jejunal digesta of birds not administered the MT. In jejunal tissue of birds not administered the MT, fructose, glycerophosphocholine, and nicotinamide adenine dinucleotide (NADH) concentrations were increased, while lysine, proline, uracil, and cytidine monophosphate concentrations were decreased.

Supervised OPLS-DA separation showed that the administration of the MT resulted in altered metabolite profiles in cecal digesta (Figure 4; Q^2^ = 0.873, *p* = 0.002; R^2^ = 0.994, *p* = 0.007). In total, 59 metabolites within cecal digesta were differentially affected by the administration of the MT (Figure 5). In this regard, metabolites involved in amino acid and short-chain fatty acid (SCFA) metabolism were conspicuously decreased in birds that did not receive the MT and were challenged with *C. perfringens*.

### 2.5. Relatively Few Viruses Were Associated with the Cecal Microbiota Propagated within Bioreactors or Birds Following Administration of the Microbiota Transplant

Metagenomic analysis of the virome revealed that assigned viral reads (family level of resolution) within the bioreactor matrix and in feces from birds ± MT administration were low, accounting for <0.025% of read counts (Appendix A). Notably, many of the prominent viruses detected have microorganisms as their primary host (e.g., *Mimiviridae*, *Myoviridae*, and *Siphoviridae*) (Appendix A). Viral families that contain members that are potential pathogens were *Herpesviridae*, *Poxviridae*, *Retroviridae*, and *Reoviridae*, but these viral families were observed at low relative levels (<0.010%). Assigned read counts of some viruses were conspicuously more abundant in birds administered the MT (i.e., *Microviridae*, *Phycodnaviridiae*, *Podoviridae*, and *Reoviridae*).

### 2.6. The Administration of a Microbiota Transplant ± Clostridium perfringens Challenge Altered Expression of Epithelium-Associated Immune Genes

Expression of hallmark genes associated with epithelial function, including barrier function, was examined at 2 and 4 days post-inoculation (p.i.) with *C. perfringens*. The mucin 2 (*MUC2B*) gene that encodes for the gel-forming mucin, MUC2, showed decreased (*p* ≤ 0.025) expression in birds challenged with *C. perfringens* (i.e., NE treatment) in comparison to Control and MT treatment birds (Figure 6A). Birds administered the MT and challenged with *C. perfringens* (i.e., MT + NE treatment) showed increased (*p* = 0.006) expression of *MUC2B* at 4-days p.i. with the pathogen in comparison to the 2-day p.i. time point (Figure 6A). Likewise, mRNA of the membrane bound mucin gene, mucin 13 (*MUC13*), was increased (*p* = 0.027) in MT + NE treatment birds in comparison to Control treatment birds (Figure 6B). mRNA of the tight junction protein gene, claudin 3 (*CLD3*) was increased (*p* < 0.001) in all birds challenged with *C. perfringens* (i.e., NE and MT + NE treatments) in comparison to birds not challenged with the pathogen and administered corticosterone (Figure 6C). Birds challenged with *C. perfringens* without MT administration exhibited decreased (*p* = 0.015) quantities of tight junction protein 1 (*TJP1*) mRNA between the 2- and 4-day p.i. time points, while birds challenged with *C. perfringens* and administered the MT showed increased (*p* = 0.007) *TJP1* mRNA between the 2- and 4-day p.i. time points (Figure 6D).

Expression of the host defence peptide (HDP) genes, cathelicidin 1 (*CATH1*) and avian β-defensin 6 (*AvBD6*), were also measured, and decreased (*p* ≤ 0.048) expression of these two HDPs were observed at 2-days p.i. in birds challenged with *C. perfringens* and administered the MT (i.e., MT + NE treatment) in comparison to Control treatment birds (Figure 6E,F). The expression of *CATH1* increased (*p* ≤ 0.043) in birds challenged with *C. perfringens* regardless of MT administration at the 4-day p.i. time point in comparison to the 2-day p.i. time point (Figure 6E). The expression of Toll-like receptor 2 family member A (*TLR2A*), interleukin 1 beta (*IL1β*), and inducible nitric oxide synthase (*INOS*), genes that are associated with epithelial inflammation were also examined. Birds challenged with *C. perfringens* exhibited decreased (*p* ≤ 0.039) expression of *TLR2A* at the 2-day but not the 4-day p.i. time point (Figure 7A). Birds challenged with *C. perfringens* but not administered the MT showed a trend for increased expression of *TLR2A*, *IL1β*, and *INOS* at the 4-day p.i. time point in comparison to the 2-day p.i. time point (Figure 7A–C), although only expression of *IL1β* was significantly altered (*p* = 0.008).

### 2.7. The Microbiota Transplant Promoted Immune Responses in the Jejunum of Birds Challenged with Clostridium perfringens

Birds administered the MT (i.e., MT and MT + NE treatments) showed increased (*p* ≤ 0.038) expression of interleukin 2 (*IL2*) in comparison to NE treatment birds (Figure 8A). Likewise, MT + NE treatment birds showed higher (*p* ≤ 0.036) expression of interleukin 17A (*IL17A*) (Figure 8B). The expression of interleukin 22 (*IL22*) was higher (*p* ≤ 0.006) in MT + NE treatment birds in comparison to all other treatments at the 2-day p.i. time point (Figure 8C). The expression of *IL22* increased (*p* = 0.021) in birds challenged with *C. perfringens* at the 4-day p.i. time point.

### 2.8. Serum Corticosterone Did Not Increase in Birds Challenged with Clostridium perfringens and Administered the Microbiota Transplant

Chickens challenged with *C. perfringens* and not administered the MT (i.e., NE treatment) exhibited a higher (*p* ≤ 0.033) serum concentration of corticosterone in comparison to Control and MT treatment birds (Appendix A). In contrast, birds challenged with *C. perfringens* and administered the MT (i.e., MT + NE treatment) did not show elevated (*p* ≥ 0.214) serum corticosterone concentrations relative to Control and MT treatment birds.

## 3. Discussion

### 3.1. Overview

Mitigation of NE using microbiota transplantation has not been explored in chickens. We observed that the administration of a MT effectively ameliorated NE in the small intestine of broilers. This included decreased necrotic lesions and reduced densities of *C. perfringens*. To date, limited research in poultry has focused on deciphering how beneficial bacteria confer disease resistance. Thus, a salient objective of the current study was to explore how the administration of a MT to day-old chicks mitigated NE by examining the intestinal microbiota, the metabolome of the enteric microbiota and host, and immune responses mounted by birds (i.e., intestinal barrier function, host defence stimulation, and impacts on mucins).

### 3.2. Enteric Bacterial Community Effects

Changes to bacterial communities in birds developing NE have been previously described [21,22]. However, it is currently unclear whether the change in the intestinal microbiota is a predisposing factor to NE, or whether it is the result of the infection [23]. Some predisposing factors to NE, such as a diet containing fishmeal, fumonisins, and/or *Eimeria* spp., have been shown to alter the intestinal microbiota [23,24]. Likewise, our previous research has shown that corticosterone administration decreases α-diversity and also alters the community structure, and concomitantly increases densities of *C. perfringens* in the small intestine of chickens [25]. Production birds lack significant maternal exposure to microorganisms, and we hypothesized that the enhancement of bacterial richness and diversity could aid in pathogen resistance. Contact with an adult hen for 24 h has been shown to increase enteric colonization by *Bacteroidetes* and *Actinobacteria* in newly hatched chicks [26]. Thus, a potential advantage of microbiota transplantation is the opportunity to increase the diversity of commensal bacteria in chicks, which may confer colonization resistance via exclusion of the pathogen from enteric niches [3]. In this regard, we showed a persistent increase in α- and β-diversity in day-old broiler birds administered a MT in the current study. Notably, we detected an increase in bacterial richness in both the jejunum and ceca of birds administered the MT. Additionally, birds administered corticosterone and challenged with *C. perfringens* displayed different compositions of bacteria with and without the MT. The increased diversity of potentially beneficial bacteria following microbiota transplantation has been previously linked to ameliorated disease in mammalian livestock species [3], which is consistent with our findings in chickens.

### 3.3. Bioreactor System to Generate the Microbiota Transplant

The MT used in the current study was generated from cecal digesta harvested from healthy adult broiler breeder males within a bioreactor system. We chose to administer bioreactor-generated microbiota (opposed to administering cecal digesta to birds) to ensure that bacteria in the MT were actively growing upon administration to birds, and because the MT propagated in a bioreactor could provide more consistency in the composition of the MT across animals (microorganisms, nutrients, etc.). Moreover, the ability to propagate the MT has logistical advantages for subsequent research using microbiota transplantation (e.g., cost in addition to quality). Similarly to a previous report [27], we observed that the diversity of bacteria within the bioreactors was reduced relative to the cecal digesta obtained from donor birds. This suggests that a restricted number of bacterial taxa are required to impart a health benefit, and this has implications for utilizing microbiota transplantation to enhance bird health, including the generation of a bacterial consortium that is free of pathogens. A salient obstacle facing fecal transplantation is the potential for inadvertent introduction of pathogenic microorganisms to the recipient [28]. Viruses are important pathogens of boilers, and we applied shotgun metagenome sequencing to examine viruses present in the MT within bioreactors, and in birds ± MT. We observed that although viruses were present within the bioreactor microbiota, and in birds administered the MT, the relative abundance of viruses was very low relative to other studies [29]. Importantly, bacteriophages are important regulators of bacterial community structures in intestines [30], and the degree to which propagation of a MT within bioreactors affects bacteriophages, as well as avian pathogenic viruses, warrants further examination.

### 3.4. Microbiota Transplantation and Enteric Bacterial Taxa

The examination of specific changes on the enteric microbiota during NE have centered around disturbances to lactic acid-producing bacteria and dietary fiber-fermenting bacteria [23]. Fishmeal diets and NE can alter the distribution of *Lactobacillus* species [21,24]. For example, in the ceca of *C. perfringens* infected birds, quantities of *Lactobacillus johnsonii* and *Lactobacillus fermentum* decreased, whereas quantities of *Lactobacillus crispatus*, *Lactobacillus pontis*, *Lactobacillus ultunese*, and *Lactobacillus salivarius* increased [21]. In the current study, *L. salivarius* was introduced with the MT and increased in abundance in birds that were administered corticosterone and challenged with *C. perfringens*. Moreover, the observed increase in abundance of *L. salivarius* corresponded with the observed reduction in *C. perfringens* densities as determined by both high-throughput sequencing and taxon-specific qPCR. Lactic acid bacteria may have played a role in reducing colonization by *C. perfringens* in the jejunum of birds administered the MT via several mechanisms. Firstly, inhibitory compounds produced by *Lactobacillus* spp. could have reduced the growth of *C. perfringens*. Reuterin and nisin, which are compounds produced by *Lactobacillus*, have been shown to inhibit the growth of vegetative cells, and inhibit the germination of *C. perfringens* endospores [31]. Secondly, *Lactobacillus*, along with other bacteria introduced with the MT, may have modulated mucosal responses that resulted in alterations to immune and barrier function that antagonized colonization by *C. perfringens* [23,32]. Thirdly, a microbial cross-feeding interaction may have occurred, as lactate produced by *Lactobacillus* spp. can be metabolized by butyrate-producing bacteria, such as *Megamonas hypermegale* [33].

### 3.5. Short-Chain Fatty Acid-Producing Bacteria

Reductions in butyrate-producing bacteria have also been demonstrated in chickens fed a diet amended with fishmeal and/or in birds infected with *C. perfringens* [21,24]. In the current study, a decrease in abundance of *M. hypermegale*, along with metabolites involved in SCFA production, was observed in the ceca of birds that were not administered the MT. As mentioned earlier, *M. hypermegale* is a butyrate-producing bacterium that is capable of converting carbohydrates into SCFAs [33]. Birds that were not administered the MT showed a decrease in butyrate, propionate, and valerate, which are prominent SCFAs produced in the ceca of birds [34]. Stimulation of SCFA production in the ceca of MT-treated birds may be one way in which the ceca microbiota was able to aid in disease resistance in the small intestine. In this regard, butyrate is an important energy source for colonocytes [35], which may confer overall health benefits. Moreover, butyrate has been shown to signal enteroendocrine cells leading to the release of glucagon-like peptide 2 into the bloodstream [23]. Notably, the systemic transport of glucagon-like peptide 2 has been demonstrated to activate nerve cells in the small intestine stimulating the release of cytokines that enhance the expression of barrier function proteins [36]. In addition, butyrate can be involved in the regulation of glucose and lipid metabolism [37]. Whether the decrease in SCFAs was due to a lower degree of fermentation of carbohydrates by the microbiota or the result of rapid uptake by the host due to increased energy demands (e.g., uptake by enterocytes), or a combination of both, warrants further examination. Regardless of specific mechanisms, the relative increase in both SCFA concentrations and the abundance of *M. hypermegale* in birds administered the MT suggests that the amelioration of disease in birds that underwent microbiota transplantation involved SCFA metabolism. Although the function of *Megamonas* spp. is not fully understood at present, in addition to production of SCFAs by bacteria within this genus, they may also provide a health benefit as a result of their ability to remove hydrogen, which may favor beneficial taxa and provide a direct health benefits to the host by improving recovery of energy from food [38].

### 3.6. Amino Acid Metabolism

Additional processes within ceca are known to confer benefits to chickens including nitrogen cycling, antibody production, and the provision of essential amino acids [38,39]. In the current study, numerous alterations to the cecal metabolome were observed in birds administered the MT that were subsequently challenged with *C. perfringens*. Notably, many amino acids were decreased in birds in which NE was manifested (i.e., in *C. perfringens*-inoculated chicks that did not receive the MT). Previous studies have shown that the branch chain amino acids (BCAAs; isoleucine, leucine, and valine) modulate the immune response of chickens [40]. We observed that all three BCAAs were down regulated in the cecal digesta in the NE treatment birds as compared to NE + MT treatment birds. This suggests that the MT may provide an increase in these amino acids, thereby enhancing immune responses resulting in amelioration of NE. Similarly, methionine is an essential amino acid that was reduced in the cecal digesta of NE treatment birds as compared to NE + MT treatment birds. Methionine is a precursor for succinyl-CoA, homocysteine, cysteine, creatine, and carnitine, and plays a key role in the immune system response of mammals [41]. Importantly, methionine is involved in the production of mucins in poultry [42]. Mucins are heavily glycosylated proteins that make up a large component of the mucus layer of the intestine, and provide protection against pathogens [43,44]. Thus, the observed reduction in methionine could be associated with decreased production of mucins, ultimately affecting the composition and quantify of mucus. This in turn could decrease the bird’s ability to protect against infection from *C. perfringens* and the manifestation of NE.

### 3.7. Mucin Responses

Additional evidence obtained in the current study indicated that the intestinal barrier of the jejunum was disrupted in birds with NE. We observed a decrease in *MUC2B* mRNA in birds challenged with *C. perfringens* that were not administered the MT. The production of mucins can be modulated by both the microbiota and immune responses [45,46]. Thus, the observed alterations to *MUC2B* mRNA expression may have corresponded to corticosterone-mediated suppression of inflammatory responses. This agrees with previous studies that demonstrated that dexamethasone exposure decreased mRNA of *MUC2* in vitro, and that corticosterone administration was implicated with decreased *MUC2B* expression in vivo [47,48]. Alternatively, the initial infection with *C. perfringens* could have led to an outflux of mucin followed by a reduction in expression, which has been observed with *Citrobacter rodentium*-mediated colonic inflammation in mice [49]. In the current study, we observed that birds administered the MT and subsequently challenged with *C. perfringens* showed positive indications of barrier restoration after infection. An increase in *MUC13* was observed 2-days p.i. with *C. perfringens*. Likewise, an increase in *MUC2B* expression was observed between the 2- and 4-days p.i. time points in birds administered the MT. Although its function has not been investigated in poultry, MUC13 is a transmembrane mucin with the potential to mediate cell signalling [50]. Studies conducted in vitro have demonstrated that MUC13 is involved in pro-inflammatory signalling, which could stimulate the production of IL8 [51]. The observed increase in *MUC13* at 2-days p.i. with *C. perfringens* and a subsequent increase in *MUC2B* at 4-days p.i. in the current study raises questions on whether *MUC13* plays a salient role in mediating mucus production during pathogen challenge. Our previous research has demonstrated that *C. perfringens* infection can modify the composition of mucus glycans [48]. Research to investigate the impacts that *C. perfringens* has on host mucins is warranted, as increases to *MUC2B* expression were only observed in birds administered the MT and subsequently challenged with the pathogen.

### 3.8. Epithelial Immune Responses

Commensal bacteria are able to promote the development of mucosal responses that limit pathogenic infections. We observed that the administration of the MT resulted in the modulation of epithelial responses that coincided with resistance to NE. In this regard, birds administered the MT and subsequently challenged with *C. perfringens* exhibited increased *TJP1* expression at 4-days p.i. with *C. perfringens*, which coincided with *MUC2B* expression. This may denote that post-infection healing responses were promoted in birds administered the MT. Importantly, an increase in *TJP1* expression as infection progressed was not observed in birds challenged with *C. perfringens* that were not administered the MT. These birds showed decreased *TJP1* expression at 4-days p.i. with *C. perfringens*, and this may indicate a disruption to barrier function. Collectively our findings implicate the restoration of intestinal barrier function with the clearance of the *C. perfringens* infection, which was only observed in birds previously administered the MT.

### 3.9. Host-Defense Peptides

Co-challenging birds with *C. perfringens* and corticosterone to incite NE modulated the expression of the HDPs, *CATH1* and *AvBD6*. Notably, these two genes showed a similar expression pattern, suggesting that they are regulated in a similar manner. At 2-days p.i. with *C. perfringens*, birds previously administered the MT showed decreased expression of *CATH1* and *AvBD6*. It has been demonstrated that *CATH1* possesses both antimicrobial and immunomodulatory properties [52]. In this regard, chicken *CATH1* has been shown to recruit neutrophils and modulate macrophage responses in a murine model. It is possible that the observed initial decrease in *CATH1* coincided with corticosterone suppression of immune defences, and this is consistent with other studies conducted in mice [53]. At 4-days p.i. with *C. perfringens*, we observed that the expression of *CATH1* increased in birds challenged with *C. perfringens*, regardless of the administration of the MT. Increased expression may have been an attempt by birds to limit *C. perfringens* infection as *AvBD6* has been previously seen to increase in the jejunum of Ross broilers with NE [54]. Alternatively, increases in HDPs could indicate their involvement in late onset immune responses to *C. perfringens* infection. This warrants further examination.

### 3.10. Epithelial Barrier and Immune-Related Function

Challenging birds with *C. perfringens* resulted in alterations to immune and barrier related function. We observed increased expression of *CLD3* in birds challenged with *C. perfringens*, which is consistent with our previous findings in white leghorn chickens administered corticosterone [48]. However, it remains unclear whether a corticosterone-mediated increase in *CLD3* expression is indicative of enhanced or reduced barrier function, as increased expression of this protein has been reported for both scenarios [55,56]. A decrease in *TLR2A* mRNA expression was observed in birds administered corticosterone and challenged with *C. perfringens* at 2-days p.i. Likewise, a previous study conducted in a layer breed showed decreased *TLR2A* expression with corticosterone administration [48]. Early hindrance to the TLR2 pathway upon pathogen challenge may be one way in which NE manifestation is facilitated in birds challenged with *C. perfringens*. A first line of defence against pathogens involves TLR signal activation of nuclear factor (NF)-κB and activating protein (AP)-1, resulting in the production of pro-inflammatory cytokines [57]. Although unknown, the administration of corticosterone in the current study may have interfered with feedback mechanisms resulting in decreased TLR2A and subsequent early immune responses. We also observed a decrease in *IL17A* mRNA in birds challenged with *C. perfringens*. Although knowledge gaps currently exist regarding the importance of Th17 responses in chickens, the importance of IL17A-mediated host defenses, such as expression of pro-inflammatory cytokine and chemokines, are recognized in other animals [58]. Collaborative induction of pro-inflammatory responses by IL17A can form a positive feedback loop that enhances the effects of IL17A [59]. Thus, it is possible that the repressive actions of corticosterone could have disrupted this feedback loop, and thereby resulted in decreased expression of IL17A in birds co-challenged with *C. perfringens* and corticosterone.

### 3.11. Beneficial Immune Responses

At the 4 day p.i. sample time, birds challenged with *C. perfringens* exhibited increased expression of *INOS*, *IL1β*, and *IL22*. This was likely due to tissue damage where inflammatory stimulation overrode the immune suppressive impacts of corticosterone administration. Significantly, birds challenged with *C. perfringens* and administered the MT did not show elevated inflammatory responses at 4-days p.i. with the pathogen, and did not develop necrotic lesions. Birds administered the MT and subsequently challenged with *C. perfringens* also exhibited stronger induction of immune responses earlier in infection. In this regard, at 2-days p.i. with *C. perfringens*, birds showed increased expression of *IL2*, *IL17A*, and *IL22* in comparison to birds challenged with the pathogen and not administered the MT. Although these cytokines are involved in inflammation, they are important mediators in mounting an immune response and limiting infection [60]. The elevation of these cytokines has been also associated with enhancement of mucosal immune development [61]. Thus, the elevated expression of *IL2*, *IL22*, and *IL17A* that was observed in the current study may be indicative of enhanced immune competence in birds as a result of the administration of the MT. This is consistent with an enhanced ability of these birds to combat infection and the observed amelioration of NE.

### 3.12. Corticosterone Levels in Blood

Free corticosterone was measured in birds to confirm that feeding corticosterone would mediate a physiological stress response. Unexpectedly, only birds challenged with *C. perfringens* and not administered the MT showed a high level of corticosterone in serum. Moreover, birds challenged with *C. perfringens* and administered the MT exhibited serum corticosterone levels equivalent to birds not administered corticosterone. Although the current study was not designed to elucidate why this occurred, several explanations are plausible. One possibility is that the endogenous production of corticosterone was increased in birds as a result of the manifestation of disease (i.e., NE treatment). Another possibility is that microorganisms introduced in the MT metabolized the corticosterone, thus leaving less corticosterone to enter systemic circulation. The breakdown of prednisolone has been demonstrated in simulated human colonic fluid [62], and several studies conducted in mammals have demonstrated that the microbiota can influence corticosterone concentrations. These observations demonstrate some possible mechanisms by which the microbiota may be involved in modulating stress responses, which requires validation in chickens.

## 4. Materials and Methods

### 4.1. Experimental Design

The factorial experiment was arranged as a complete randomized design with four treatments with two levels of MT administration (±MT), and two levels of pathogen administration (± *C. perfringens* and corticosterone administration). The four experimental treatments were: (i) the Control treatment (birds not administered the MT and not challenged with *C. perfringens*); (ii) the MT treatment (birds administered the MT and not challenged with *C. perfringens*); (iii) the NE treatment (birds not administered the MT and challenged with *C. perfringens*); and (iv) the MT + NE treatment (birds administered the MT and challenged with *C. perfringens*). Treatment iii and iv birds were administered corticosterone in their diet. Birds from each treatment were euthanized at two time points (i.e., 2- and 4-days p.i. with *C. perfringens*), and there were three replicate birds per treatment-time combination. It is noteworthy that care was taken to ensure that the replicates were independent (e.g., separately prepared *C. perfringens* and MT inocula).

### 4.2. Animals and Husbandry

Ross 308FF broiler eggs were obtained from a local hatchery (Lethbridge, AB, Canada) and were incubated in a Brinsea Ovation 56 EX fully automatic digital egg incubator (Brinsea Products Inc., Titusville, FL, USA) as previously described [20]. After hatch, arbitrarily chosen chicks were placed within separate individually ventilated cages (1862 cm^2^ floor space; Techniplast, Montreal, QC, Canada) with sterile wood shavings for bedding. Birds were provided free access to water and food at all times. A starter diet was provided for the first 10 days post-hatch (Appendix A). Commencing on day 11 post-hatch until the end of the experiment, a grower diet was introduced. Fresh feed was provided to birds each morning and afternoon, and birds were transferred to clean cages daily. Birds were maintained at 30 °C for the first 2 days post-hatch, 28 °C for the next 2 days, then maintained at 26 °C for the remainder of the experiment. Birds were maintained on an 18 h light: 6 h dark cycle throughout the experimental period.

### 4.3. Microbiota Transplant Generation

Generation of the MT was conducted as described previously [27]. Briefly, digesta from the ceca of six-month-old healthy male broiler breeders was harvested within 30 min of death. Ceca were ligated at the ileal-cecal junctions to prevent infiltration of air into ceca, and excised from the intestinal tract. The ligated ceca were immediately transferred into a Thermo Forma 1025 anaerobic chamber (Thermo Fisher Scientific Inc., Waltham, MA, USA) containing a nitrogen-predominant atmosphere (85:5:10% N_2_:H_2_:CO_2_). Digesta was removed from the ceca and thoroughly mixed, transferred to tubes, and stored at −80 °C until required. A continuous-flow mini-bioreactor system was used to propagate the MT [62]. Mini-bioreactors were situated within a dedicated Thermo Forma 1025 anaerobic chamber (Thermo Fisher Scientific Inc.) containing a nitrogen-predominant gas atmosphere with a 37 °C ambient temperature generated with thermostat controlled heaters affixed to the roof of the anaerobic chamber. Within the anaerobic chamber, cecal digesta was thawed on ice, and suspended in a reduced medium consisting of 1 g/L tryptone, 2 g/L proteose peptone, 2 g/L yeast extract, 0.1 g/L arabinogalactan, 0.15 g/L maltose, 0.15 g L D-cellobiose, 0.4 g/L sodium chloride, 5 mg/L hemin, 0.01 g/L magnesium sulfate, 0.01 g L calcium chloride, 0.04 g/L potassium phosphate monobasic, 0.04 g/L potassium phosphate dibasic, and 2 mL/L Tween 80 at pH 6.8 [63]. The diluted cecal digesta was added to individual bioreactor vessels at a final concentration of 25% (*w*/*v*) and allowed to incubate in the medium for 1 day without continuous flow of media. Each bioreactor vessel had an inflow and outflow tube that connected to a peristaltic pump that facilitated nutrient exchange (3.76 mL/h). After 10 days, the MT was collected, and individual chicks (1-day old) were gavaged with 1 mL of the MT or medium alone using an 18-gauge, 5-cm-long gavage needle. In addition, a 1 mL sample from each bioreactor was centrifuged at 13,000× *g* for 5 min, the supernatant was removed, and the pellet was stored at −80 °C until processing for bacterial community analysis by 16S rRNA gene sequencing. Samples were also stored for virome metagenome analysis.

### 4.4. Necrotic Enteritis Model

To induce a controlled physiological stress response, NE and MT + NE treatment birds were administered corticosterone in feed (20 mg/kg) beginning at 11 days post-hatch as previously described [20]. It is noteworthy that the corticosterone stress-predisposition model results in a high prevalence of acute enteritis. Corticosterone was not incorporated into the feed of Control and MT treatment birds. At days 12 and 13 post-hatch, birds were orally inoculated with 1 mL of *C. perfringens* containing 1–2 × 10^8^ colony forming units (i.e., NE and MT + NE treatment birds) or 1 mL of medium alone (i.e., Control and MT treatment birds) using an 18-gauge gavage needle as described above for administration of the MT. Briefly, inoculum was generated by growing a starter culture of a virulent Type G strain of *C. perfringens* (CP1) [64,65] in heart infusion broth at 37 °C within a Thermo Forma 1025 anaerobic chamber containing a nitrogen-predominant atmosphere. The following day, 2.5 mL of the culture was transferred to 50 mL fluid thioglycolate medium within the anaerobic chamber, and the culture was incubated at 37 °C for 4 h before inoculation of birds.

### 4.5. Collection of Feces

For characterization of the virome, fresh feces was collected from birds 9 days after administration of the MT. Immediately after collection, individual fecal samples were snap frozen in liquid nitrogen, and stored at −80 °C until processed.

### 4.6. Animal Euthanasia and Sample Collection

At the two defined endpoints, birds were anesthetized with isoflurane (5% isoflurane; 1 L O_2_/min) and humanely euthanized by cervical dislocation while under general anesthesia. Immediately after euthanization, the abdomen was opened, and blood was drawn directly from the heart, and serum was collected for corticosterone quantification. The small intestine was aseptically removed, incised, photographed, examined for disease progression, and lesions were scored (see below). Within 5 min of euthanization, jejunal tissue was flash frozen in liquid nitrogen for metabolomic analysis, stored in RNAlater™ Stabilization Solution (Thermo Fisher Scientific Inc.) for quantification of mRNA, and placed in 10% neutral buffered formalin (Leica, Concord, ON, Canada) for histopathological examination. Digesta from the jejunum and ceca was removed using a sterile wooden splint, and snap frozen. With the exception of samples for histopathologic analysis, all samples were stored at −80 °C until processed.

### 4.7. Lesion Scoring and Histopathology

The entire length of the small intestine (duodenum to ileal-cecal junction) was examined for gross lesions and scored on a scale of 0 to 6 as described by Shojadoost et al. [66]. Samples for histopathologic examination were processed as described previously [20]. Briefly, the proximal jejunum was examined, and care was taken to ensure that lesions were not sampled. Jejunal tissue was fixed in formaldehyde for a minimum of 24 h. Fixed samples were embedded in paraffin blocks, sectioned (5 µm), de-paraffinized with xylene, and stained with hematoxylin and eosin [48]. Sections were scored by a pathologist (V.F.B.) who was blinded to treatments, using a modified scoring system based on a previously described scoring systems [67,68,69]. In this regard, sections were graded from 0 to 4 for villar fusion, villar atrophy, mucosal necrosis, bacterial invasion, and changes to the lamina propria that ranged from mild edema to coagulative necrosis [20]. Total histologic scores were determined by calculating the sum of scores from all categories. Data were analyzed using the Kruskal–Wallis test with Dunn’s test for multiple comparison (Version 9.1.2, GraphPad Prism; La Jolla, CA, USA). *p*-values ≤ 0.050 were considered significant.

### 4.8. Quantification of Clostridium perfringens

DNA was extracted from jejunal digesta using a Qiagen QIAamp DNA Fast Stool Kit (Qiagen Inc., Toronto, ON, Canada) with modifications described previously [20]. A standard curve of known copies of 16S ribosomal RNA (rRNA) gene DNA specific to *C. perfringens* was generated as previously described [25,48]; the standard curve was generated using a 10-fold dilution series ranging from 10^1^ to 10^7^ copies of the *C. perfringens* 16S rRNA gene. Primers for *C. perfringens* were F: 5′-AAAGATGGCATCATCATTCAAC and R: 5′-TACCGTCATTATCTTCCCCAAA (Integrated DNA Technologies, Coralville, IA, USA) [70]. Each PCR reaction contained 10 μL of Quantitect SYBR green master mix (Qiagen Inc.), 1 μL of each primer, 2 μL of bovine serum albumin, 4 μL of DNase-free water, and 2 μL of template DNA (10 ng/μL). Reaction conditions were: 95 °C for 15 min; and 40 cycles of 95 °C for 15 s, 55 °C for 30 s, and 72 °C for 30 s, followed by melt curve analysis from 55 to 95 °C. An Mx3005p thermocycler (Agilent Technologies, Santa Clara, CA, USA) was used. Each reaction was run in duplicate, and the mean of the two observations was calculated. Data was log transformed to achieve normality, and analyzed by two-way analysis of variance (ANOVA) using GraphPad Prism. In conjunction with a significant main effect, means were compared using a least significant difference test. *p*-values ≤ 0.050 were considered significant.

### 4.9. Characterization of Bacterial Communities

Jejunal and cecal digesta DNA was extracted using the Qiagen QIAamp DNA Fast Stool Kit (Qiagen Inc.) as described previously [20]. Briefly, the Variable 4 (V4) region of the 16S rRNA gene was amplified using a protocol developed by Kozich et al. [71]; the primers used were F-5′-GTGCCAGCMGCCGCGGTAA-3′ and R-5′-GGACTACHVGGGTWTCTAAT-3′. Each PCR reaction contained: 12.5 μL of Paq5000 Hi Fidelity Taq Master Mix (Agilent Technologies Canada Inc., Mississauga, ON, Canada), 1 μL each of primer (10 μM; Integrated DNA Technologies), 5 μL (jejunum) or 2 μL (ceca) of DNA, and 5.5 μL (jejunum) or 8.5 μL (ceca) of nuclease-free water (Qiagen Inc.). Reaction conditions were: 95 °C for 2 min; 25 (ceca) or 30 (jejunum) cycles of 95 °C for 20 s, 55 °C for 15 s, and 72 °C for 5 min; and 1 cycle at 72 °C for 10 min. AMPure XP beads (Beckman Coulter Diagnostics, Brea, CA, USA) were used to purify PCR amplicons. Agilent High Sensitivity DNA chips were used on a Bioanalyzer 2100 (Agilent Technologies Canada Inc.) to check the purified amplicons for quality and size. Amplicons were quantified using a Qubit 4 fluorometer (Thermo Fisher Scientific Inc.), DNA samples were normalized to 6 nM, pooled, denatured with sodium hydroxide, and diluted with HT1 (Illumina Inc., San Diego, CA, USA) to produce a 6 pM library for sequencing analysis. PhiX control DNA (Illumina Inc.; 25%) was added to the library as a sequencing control. The library was loaded onto a MiSeq Reagent Kit v2 500-cycle and run on an Illumina MiSeq platform (Illumina Inc.).

Quantitative Insights Into Microbial Ecology 2 (QIIME2™, version 2021.2) [72] was used to execute sequencing analysis. DADA2 was used to filter low quality reads (quality score < 20) and trim sequences. Forward reads were grouped into exact ASVs, and taxonomy was classified using the SILVA bacteria reference database (release 138) [73]. Low count reads, mitochondrial sequences, and chloroplast sequences were removed. The minimum sampling depth was 11,012 and 85,403 reads for the jejunal and cecal digesta, respectively. There was no difference among the two sample time points, and the analysis was averaged over the two times. Core metrics phylogeny analysis was performed in QIIME2 to obtain α- and β-diversity. Alpha diversity was analyzed in QIIME2 by pairwise comparisons of Kruskal–Wallis test. Beta diversity was analyzed by pairwise permutational multivariate analysis of variance (PERMANOVA) in QIIME2 [74]. A Bejamini and Hochberg correction was applied to both α- and β-diversity tests when corrected *p*-values were ≤0.050. Percent abundance of taxa (e.g., *Clostridium sensu stricto* 1 and *Lactobacillus* spp. in the jejunum, and *Megamonas* spp. and *Bacteroides* spp. in the ceca) was assessed for normality. Two-way ANOVA was applied to normally distributed data, whereas Kruskal–Wallis test was applied to data that failed to achieve normality. *p*-values ≤ 0.050 were considered significant.

### 4.10. Characterization of the Virome

DNA and RNA viruses in fecal samples obtained from birds (2 g) and from MT bioreactors (2 mL) were examined using metagenome sequence analysis as described previously [29]. Viral abundance at a family level of resolution was calculated relative to total microbial read counts (i.e., eukaryotes, prokaryotes, and viruses).

### 4.11. Characterization of the Metabolome

A 700 MHz NMR spectrometer (Bruker Avance III HD NMR spectrometer; Bruker, Milton, ON, Canada) was used to collect the metabolomics data as described previously [34]. Briefly, jejunal digesta, jejunal tissue, and cecal digesta samples (150 mg) were suspended in metabolomics buffer (0.125 M KH_2_PO_4_, 0.5 M K_2_HPO_4_, 0.00375 M NaN_3_, and 0.375 M KF; pH 7.4). Samples were homogenized using one 6-mm-diameter steel bead for 10 min using a Qiagen Tissue Lyser LT (Qiagen Inc.) operated at 50 Hz. Homogenized samples were centrifuged for 5 min at 14,000× *g*. The supernatant of each sample was passed through a 3000 MWCO Amicon Ultra-0.5 filter (Millipore Sigma, Oakville, ON, Canada) by centrifuging at 14,000× *g* for 30 min at 4 °C; filters were rinsed with Millipore water ten times prior to use. For each filtrate, 360 μL was mixed with 200 μL metabolomics buffer and 140 μL deuterium oxide containing 0.05% *v*/*v* trimethylsilylpropanoic acid (TMSP) to yield a final volume of 700 μL. TMSP was used as a chemical shift reference for ^1^H-NMR spectroscopy. To prepare the solution for spectroscopy, it was vortexed, centrifuged at 12,000× *g* for 5 min at 4 °C, and a 550 μL aliquot of the supernatant was loaded in a 5-mm NMR tube. NMR spectra were acquired, and both the data collection and spectral processing were conducted as described previously [34]. MATLAB (MathWorks, Natick, MA, USA) was used to align spectral peaks through Recursive Segment Wise Peak Alignment [75] and binning with Dynamic Adaptive Binning [76]. Data were normalized to the total metabolome, excluding the region containing the water peak, and pareto scaled.

The two time points p.i. were combined, and MATLAB (Math Works) and the Metaboanalyst R package was used for analysis [77]. To determine which metabolites were significantly altered between treatments, spectral bins were subjected to univariate and multivariate analysis. The univariate measure was calculated in MATLAB using a decision tree algorithm, as previously described by Goodpaster et al. [78]. The multivariate tests utilized the Variable Importance Analysis based on the random Variable Combination (VIAVC) algorithm. This combines both Partial Least Squares Discriminant Analysis (PLS-DA) and the area under the Receiver Operating Characteristics (ROC) curve to synergistically determine the best subset of metabolites for group classifications [79]. All *p*-values obtained from the analysis were Bonferroni–Holm corrected for multiple comparisons. MATLAB was used to calculate the percent difference of bins between treatments. Metaboanalyst R package was used to facilitate orthogonal partial least squares discriminant analysis (OPLS-DA). Chenomx 8.2 NMR Suite (Chenomx Inc., Edmonton, AB, Canada) was used to identify metabolites, and MetaboAnalyst 5.0 was used to determine metabolic pathways that were associated with relevant metabolites [80]. *p*-values ≤ 0.050 were considered significant.

### 4.12. Quantification of mRNA

RNA was extracted from jejunal tissue using RNeasy Plus Mini Kit (Qiagen Inc.) and assessed for quality and quantity using an Agilent Bioanalyzer (Agilent Technologies). RNA (1 μg) was reverse transcribed to cDNA using a QuantiTect reverse transcription kit (Qiagen Inc.). A Mx3005p thermocycler (Agilent Technologies) was used to perform PCR reactions. Each PCR reaction contained 5 μL of QuantiTect SYBR green master mix (Qiagen Inc.), 0.5 μL of each primer (10 μM), 3 μL of RNase-free water, and 1 μL of cDNA. Reaction conditions were: 95 °C for 15 min; 40 cycles of 95 °C for 15 s, 55–58 °C for 30 s, and 72 °C for 30 s; followed by melt curve analysis from 55 to 95 °C. Unless cited otherwise, primers were designed using National Center for Biotechnology Information (NCBI) primer basic local alignment search tool (Appendix A). Reactions were run in duplicate, and the mean of the two observations was used to calculate mRNA concentrations relative to the reference genes, β-actin and glyceraldehyde 3-phosphate dehydrogenase (GAPDH) using qBase+ software (Biogazelle, Gent, Belgium) [81]. Data was log transformed to achieve normality, and analyzed by two-way ANOVA. In conjunction with a significant main effect, means were compared using a least significant difference test. *p*-values ≤ 0.050 were considered significant.

### 4.13. Quantification of Corticosterone Serum Concentrations

Serum corticosterone concentrations were measured by enzyme-linked immunosorbent assay (Cayman Chemical Company, Ann Arbor, MI, USA) following the manufacturer’s recommendations. Concentrations were computed using a logistic regression model, and data were analyzed by two-way ANOVA. In conjunction with a significant main effect, means were compared using a least significant difference test. *p*-values ≤ 0.050 were considered significant.

## 5. Conclusions

A single administration of a MT to day-old chicks ameliorated stress-induced NE. Despite a loss in diversity of bacteria propagated within bioreactors, increased α-diversity and altered β-diversity of bacterial communities in both the jejunum and ceca of the MT-treated birds were observed. Furthermore, the metabolome of cecal digesta, and to a lesser extent of jejunal digesta and tissue, showed a significant number of altered metabolites in birds administered the MT. This included an increase in SCFA-producing bacteria, as well as relative concentrations of SCFAs, BCAAs, and methionine in cecal digesta. These metabolites have important roles in the immune response, cell signalling, and mucosal integrity of the intestine. This suggests that the cecal microbiota imparts a systemic influence on resistance to NE. Corticosterone-induced stress was observed to promulgate disease via the impairment of immune and barrier function, and the administration of the MT counteracted these impacts by promoting mucosal immune competence. Collectively, our results demonstrate the importance of exposing young birds to a diverse set of microorganisms soon after hatch. Artificially increasing microbial diversity early in life may aid in the promotion of positive microorganism-microorganism and microorganism-host interactions that promote immune development and resistance to important diseases, including NE. Strategies to cost-effectively apply a microbiota that confers a health benefit (e.g., administration on eggs), and validation of efficacy in production settings is required. It is anticipated that the finds from the current study will facilitate studies to address these topics.

## Figures and Tables

**Figure 1 pathogens-11-00972-f001:**
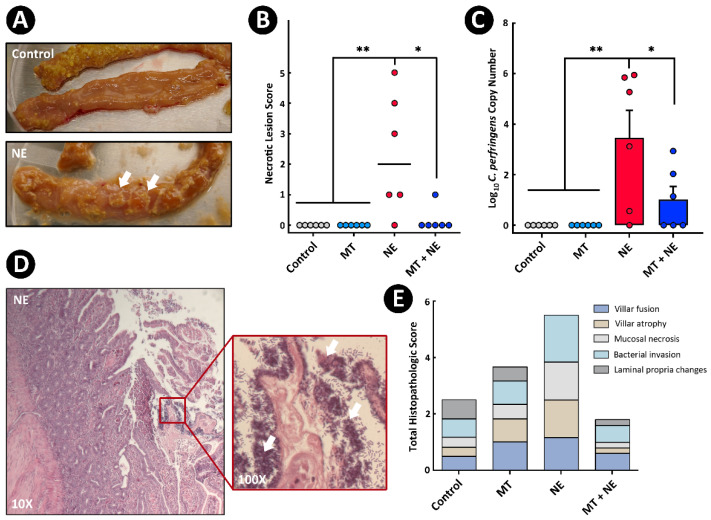
Gross pathologies, histopathologic changes, and quantitative PCR for *Clostridium perfringens* in the jejunum. At 1-day post-hatch, birds were orally administered a microbiota transplant (MT) originating from adult broiler breeder birds and propagated within bioreactors. On days 12 and 13 post-hatch, birds were orally administered 1–2 × 10^8^ colony forming units of *C. perfringens*, the incitant of necrotic enteritis (NE) (i.e., NE and MT + NE treatments) or buffer alone (i.e., Control and MT treatments). (**A**) Longitudinally incised jejunum showing fibrin development in NE treatment (arrows) relative to a healthy intestine (i.e., Control treatment). (**B**) Necrotic lesion scoring. (**C**) Densities of the *C. perfringens* 16S rRNA gene in jejunum digesta as measured by qPCR. (**D**) Micrographs showing bacteria in association with jejunal mucosa (arrows). (**E**) Stacked bar plot showing total histopathologic scores by metric. Six replicate birds were analyzed per treatment. * *p* ≤ 0.050. ** *p* ≤ 0.010.

**Figure 2 pathogens-11-00972-f002:**
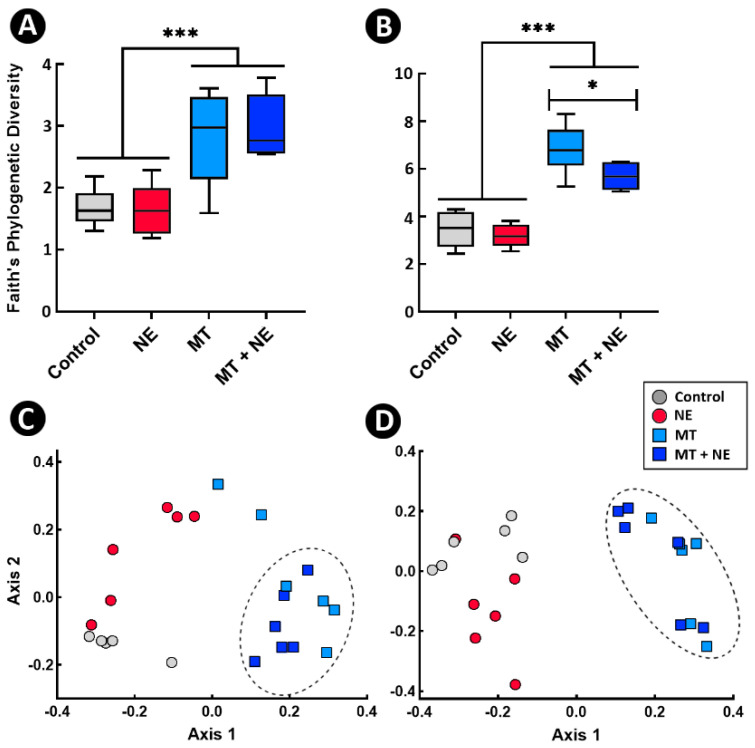
Bacterial diversity within jejunal and cecal digesta. At 1-day post-hatch, birds were orally administered a microbiota transplant (MT) originating from adult broiler breeder birds and propagated within bioreactors. On days 12 and 13 post-hatch, birds were orally administered 1–2 × 10^8^ colony forming units of *Clostridium perfringens*, the incitant of necrotic enteritis (NE) (i.e., NE and MT + NE treatments) or buffer alone (i.e., Control and MT treatments). (**A**,**B**) Faith’s phylogenetic diversity within the jejunum (**A**) and ceca (**B**). (**C**,**D**) Unweighted UniFrac principal coordinate analysis plot of β-diversity in the jejunum (**C**) and ceca (**D**). Bacterial community structure between birds ± MT treatment in jejunum and ceca differed (*p* = 0.001). Six replicate birds were analyzed per treatment.* *p* < 0.050 and *** *p* < 0.001.

**Figure 3 pathogens-11-00972-f003:**
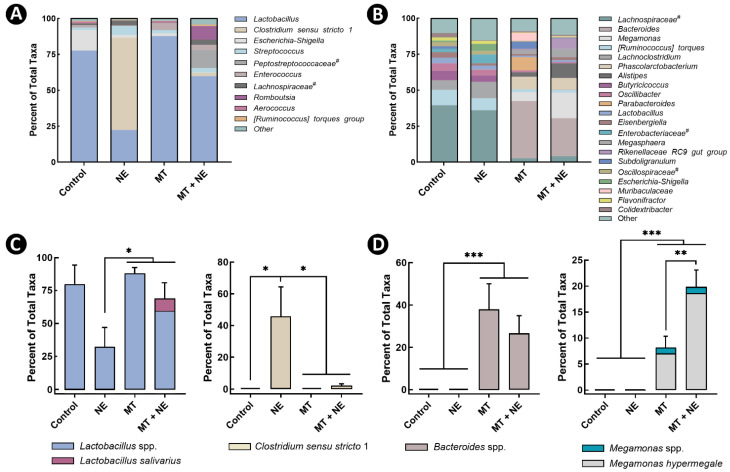
Composition of the microbiota within jejunal and cecal digesta. At 1-day post-hatch, birds were orally administered a microbiota transplant (MT) originating from adult broiler breeder birds and propagated within bioreactors. On days 12 and 13 post-hatch, birds were orally administered 1–2 × 10^8^ colony forming units of *Clostridium perfringens*, the incitant of necrotic enteritis (NE) (i.e., NE and MT + NE treatments) or buffer alone (i.e., Control and MT treatments). (**A**) Percent abundance of total taxa in jejunal digesta, with taxa that comprised >1% of total abundance shown. (**B**) Percent abundance of total taxa in the cecal digesta, with taxa that comprised >3% of total abundance shown. (**C**) Percent abundance of *Lactobacillus* spp. and *Clostridium sensu stricto 1* (including *C. perfringens*) in jejunal digesta. (**D**) Percent abundance of *Bacteroides* spp. and *Megamonas* spp. in cecal digesta. # denotes taxa delineated at the family level. Six replicate birds were analyzed per treatment. * *p* < 0.050, ** *p* < 0.010, and *** *p* < 0.001.

**Figure 4 pathogens-11-00972-f004:**
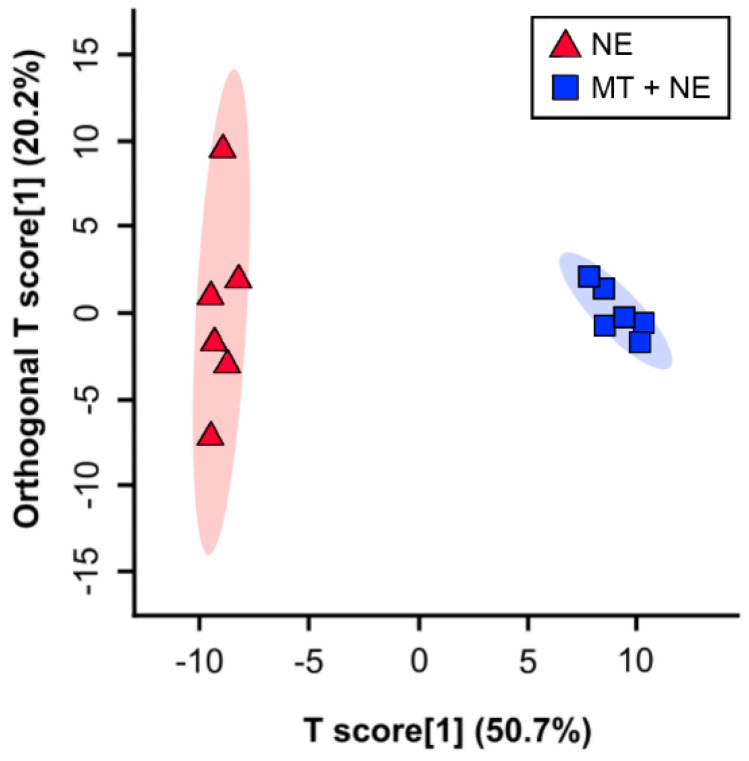
Orthogonal Projections to Latent Structures Discriminant Analysis (OPLS-DA) scores plot of metabolites within cecal digesta. At 1-day post-hatch, birds were orally administered a microbiota transplant (MT) originating from adult broiler breeder birds and propagated within bioreactors. On days 12 and 13 post-hatch, birds were orally administered 1–2 × 10^8^ colony forming units of *Clostridium perfringens*, the incitant of necrotic enteritis (NE) (i.e., NE and MT + NE treatments). Each triangle or square represents one bird, and data were plotted using metabolites identified to be significant by a Mann–Whitney U test and/or Variable Importance Analysis based on Variable Combination (VIAVC) machine learning. Cross-validation of the OPLS-DA model for the cecal digesta provided an excellent model fit (Q^2^ = 0.873, *p* = 0.002) and explained most of the variance (R^2^ = 0.994, *p* = 0.007), indicating that the multivariate difference observed between the NE and MT + NE treatments was real. Six replicate birds were analyzed per treatment.

**Figure 5 pathogens-11-00972-f005:**
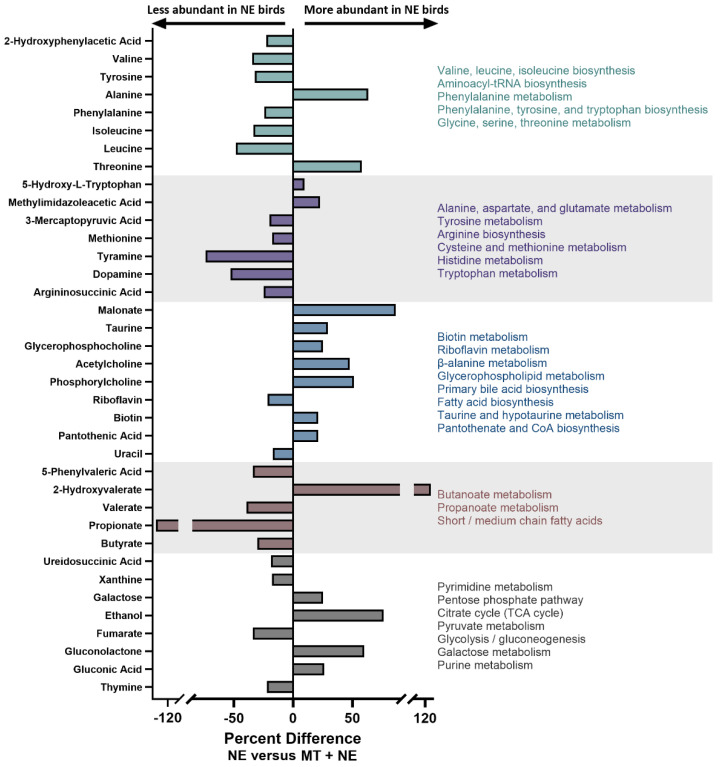
Metabolite profiles and associated functional pathways within cecal digesta. At 1-day post-hatch, birds were orally administered a microbiota transplant (MT) originating from adult broiler breeder birds and propagated within bioreactors. On days 12 and 13 post-hatch, birds were orally administered 1–2 × 10^8^ colony forming units of *Clostridium perfringens*, the incitant of necrotic enteritis (NE) (i.e., NE and MT + NE treatments). Metabolites that were differentially abundant between NE treatment and MT + NE treatment birds are shown. Six replicate birds were analyzed per treatment.

**Figure 6 pathogens-11-00972-f006:**
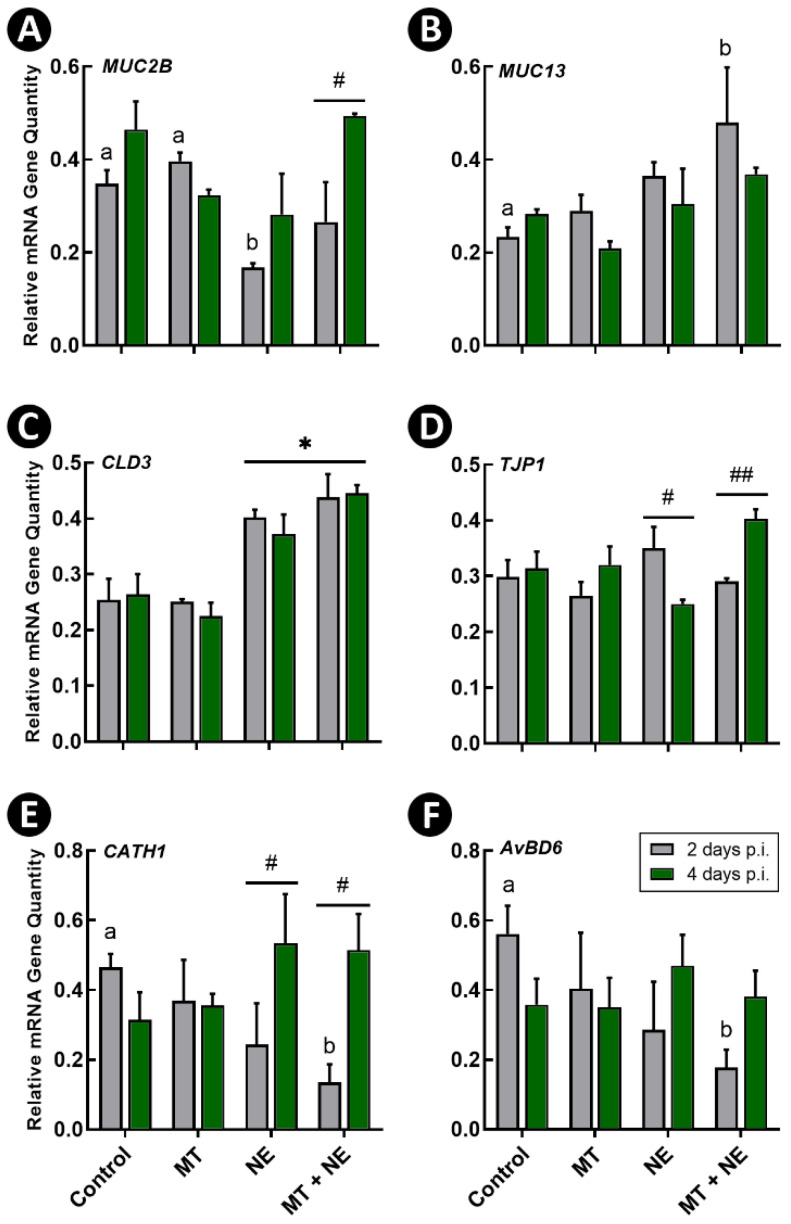
Relative mRNA gene quantities in jejunal tissue 2- and 4-days post-inoculation of birds with *Clostridium perfringens*, the incitant of necrotic enteritis (NE). At 1-day post-hatch, birds were orally administered a microbiota transplant (MT) originating from adult broiler breeder birds and propagated within bioreactors. On days 12 and 13 post-hatch, birds were orally administered 1–2 × 10^8^ colony forming units of *C. perfringens* (i.e., NE and MT + NE treatments) or buffer alone (i.e., Control and MT treatments). (**A**) *MUC2B.* (**B**) *MUC13.* (**C**) *CLD3.* (**D**) *TJP1.* (**E**) *CATH1.* (**F**) *AvBD6*. Three replicate birds were analyzed per treatment and time. Histogram bars denoted with different letters differ (*p* < 0.050) among treatments for the 2-day post-inoculation (p.i.) time point. The asterisk indicates a difference between NE and non-NE treatments (* *p* < 0.001). # denotes a difference between the 2- and 4-day p.i. time points within the same treatment at *p* < 0.050, and ## denotes a difference at *p* < 0.010.

**Figure 7 pathogens-11-00972-f007:**
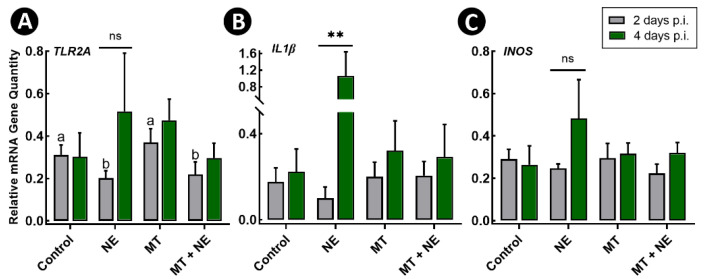
Relative mRNA gene quantities in jejunal tissue 2- and 4-days post-inoculation (p.i.) of birds with *Clostridium perfringens*, the incitant of necrotic enteritis (NE). At 1-day post-hatch, birds were orally administered a microbiota transplant (MT) originating from adult broiler breeder birds and propagated within bioreactors. On days 12 and 13 post-hatch, birds were orally administered 1–2 × 10^8^ colony forming units of *C. perfringens* (i.e., NE and MT + NE treatments) or buffer alone (i.e., Control and MT treatments). (**A**) *TLR2A*; (**B**) *IL1β*; and (**C**) *INOS*. Three replicate birds were analyzed per treatment and time. Histogram bars denoted with different letters differ (*p* < 0.050) among treatments for the 2-day p.i. time point. Double asterisks denote a difference between the 2- and 4-day p.i. time points within the same treatment (*p* < 0.010). ns denotes “not significant”.

**Figure 8 pathogens-11-00972-f008:**
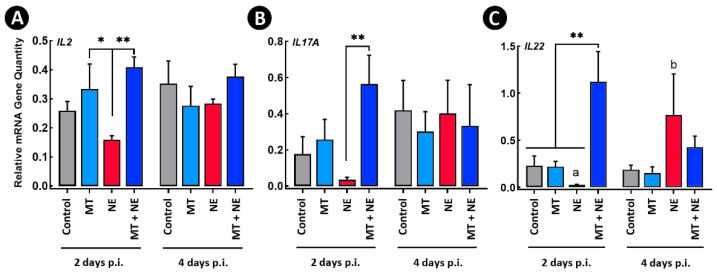
Relative mRNA gene quantities in jejunal tissue 2- and 4-days post-inoculation of birds with *Clostridium perfringens*, the incitant of necrotic enteritis (NE). At 1-day post-hatch, birds were orally administered a microbiota transplant (MT) originating from adult broiler breeder birds and propagated within bioreactors. On days 12 and 13 post-hatch, birds were orally administered 1–2 × 10^8^ colony forming units of *C. perfringens* (i.e., NE and MT + NE treatments) or buffer alone (i.e., Control and MT treatments). (**A**) *IL2*. (**B**) *IL17A*. (**C**) *IL22*. Three replicate birds were analyzed per treatment and time. Histogram bars denoted with different letters differ (*p* < 0.050) between the two time points among the same treatment. Asterisks indicate treatments that differ within individual time points, where * is *p* < 0.050, and ** is *p* < 0.010.

## Data Availability

The microbiota raw sequencing reads were submitted to the Sequencing Read Archive of NCBI under BioProject accession number PRJNA835063.

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
