# Peer review of "Microbiota Transplantation in Day-Old Broiler Chickens Ameliorates Necrotic Enteritis via Modulation of the Intestinal Microbiota and Host Immune Responses"

_pathogens, 2022, doi:10.3390/pathogens11090972_

Round 1

Reviewer 1 Report

Dear Authors, I would like to submit just 4 points to review and share. In any case, congratulations on the excellent work:

Writing mistakes:

1) ganisms . Chickens populate their gastro-intestinal tract via ingestion of microorganisms 76

2) func tion, was examined at 2 and 4 days post-inoculation (p.i.) with C. perfringens. The 242

More to report:

3) 2.2. Administration of the Microbiota Transplant Reduced Pathologies.....it might be useful to enter the number of animals per group (I suppose 6 based on the figures). Likewise the number of animals (total in this case) could be entered at the point 4.2. Animals and Husbandry

4) we observed that the diversity of bacteria within the bioreactors was reduced relative to the cecal digesta obtained from donor birds. This suggests that a restricted number of bacterial taxa is required to impart a health benefit, and this has implications for utilizing microbiota transplantation to enhance bird health, including the generation of a bacterial consortium that is free of pathogens.....I agree but the selection could be generated by in vitro cultivation and by the limits introduced by the fermentation process. In any case, it is true that the results confirm what you have said.

Overall I would like to reiterate that in my opinion the work is extremely interesting, well set up and well done; a great deal of data has been collected and many important results have been achieved.

Author Response

Comments and Suggestions for Authors

Reviewer comment. Dear Authors, I would like to submit just 4 points to review and share. In any case, congratulations on the excellent work:

Writing mistakes:

1) ganisms . Chickens populate their gastro-intestinal tract via ingestion of microorganisms 76

2) func tion, was examined at 2 and 4 days post-inoculation (p.i.) with C. perfringens. The 242

Author response. Micro-organisms (L75 and L76) was automatically hyphenated by the Pathogens manuscript template.  The wayward space in func_tion has been corrected.

More to report:

Reviewer comment. 3) 2.2. Administration of the Microbiota Transplant Reduced Pathologies.....it might be useful to enter the number of animals per group (I suppose 6 based on the figures). Likewise the number of animals (total in this case) could be entered at the point 4.2. Animals and Husbandry

Author response. The number of replicate birds has been added.

Reviewer comment. 4) we observed that the diversity of bacteria within the bioreactors was reduced relative to the cecal digesta obtained from donor birds. This suggests that a restricted number of bacterial taxa is required to impart a health benefit, and this has implications for utilizing microbiota transplantation to enhance bird health, including the generation of a bacterial consortium that is free of pathogens.....I agree but the selection could be generated by in vitro cultivation and by the limits introduced by the fermentation process. In any case, it is true that the results confirm what you have said.

Author response. We agree, and this is something we are currently addressing in a separate study.

Reviewer comment. Overall I would like to reiterate that in my opinion the work is extremely interesting, well set up and well done; a great deal of data has been collected and many important results have been achieved.

Author response. We thank the reviewer for the positive comments.

Reviewer 2 Report

The manuscript "Microbiota Transplantation in Day-Old Broiler Chickens Ameliorates Necrotic Enteritis (NE) via Modulation of the Intestinal Microbiota and Host Immune Responses" by Zaytsoff et al describes the beneficial effects of microbiota transplant (MT) to broilers in reducing NE severity. The manuscript is very well written and the findings are very novel and interesting with the conclusions carrying a good merit. However, I have two major comments/ concerns that have to be addressed and a few minor suggestions/corrections.

1. The crux of this study is the NE lesions/pathology and form the figure 1B, it looks like they examined/necropsied only 6 birds per group? If yes, they need to explain why such a small number? One would ideally need at least 12-15birds per group to allow a good NE outcome readout. It's such a heterogeneous outbred population and you can already see such a wide variation amongst birds showing lesions with 1 bird showing 0 lesions, while others are scattered all over the place. If you use some 15-20 birds, you will obviously increase the stringency or confidence of you findings. It is very fine if you sample only 6 birds for other analysis such as microbial, cecal or gene expression etc but for gross pathology, one has to have more birds to depict the findings with high level of confidence. This point may diminish the substance of their findings.

Which stat was used for lesion scores analysis? The most appropriate is Fisher's exact test (2x2 contingency table). I am not sure how you can see a significance with just 6 biological reps.

Authors also need to mention if this model simulates a 'subclinical' or a 'clinical' NE. Based on the average lesion score, it looks mild but again it's very difficult since only 1 bird had 5 score. Please mention the source of CP1 and looks like it has a 'low' level of virulence in vivo.

Authors need to mention how many birds per treatment used? Not sure why I wasn't able to find it in the M&M design.

2. Page 337: NE amelioration was observed only in jejunum? We always find lesions in duodenum and/or jejunum under experimental NE setting. Page 358-359: The premise that MT can displace pathogenic C. perfringens MAY be a little flawed since certain CP strains carry signature virulence genes that are absent in commensal CP and such virulent strains don't usually reside as opportunistic CP but gain entry upon intestinal predisposition.

Author Response

Comments and Suggestions for Authors

Reviewer comment. The manuscript "Microbiota Transplantation in Day-Old Broiler Chickens Ameliorates Necrotic Enteritis (NE) via Modulation of the Intestinal Microbiota and Host Immune Responses" by Zaytsoff et al describes the beneficial effects of microbiota transplant (MT) to broilers in reducing NE severity. The manuscript is very well written and the findings are very novel and interesting with the conclusions carrying a good merit. However, I have two major comments/ concerns that have to be addressed and a few minor suggestions/corrections.

  1. The crux of this study is the NE lesions/pathology and form the figure 1B, it looks like they examined/necropsied only 6 birds per group? If yes, they need to explain why such a small number? One would ideally need at least 12-15birds per group to allow a good NE outcome readout. It's such a heterogeneous outbred population and you can already see such a wide variation amongst birds showing lesions with 1 bird showing 0 lesions, while others are scattered all over the place. If you use some 15-20 birds, you will obviously increase the stringency or confidence of you findings. It is very fine if you sample only 6 birds for other analysis such as microbial, cecal or gene expression etc but for gross pathology, one has to have more birds to depict the findings with high level of confidence. This point may diminish the substance of their findings.

Author response. The reviewer’s comment is well taken, and more birds would have been better. Of note, we used a new model of acute necrotic enteritis, which we have recently published [1], and this reference is cited in the manuscript. Importantly, this model is a high prevalence model of acute necrotic enteritis. Moreover, our institutional Animal Care Committee limited the number of birds that we could include in the experiment. Importantly, we do not make any statements that microbiota transplantation is ready to be transferred to the sector. Rather we state that,

“Collectively, our results demonstrate the importance of exposing young birds to a diverse set of microorganisms soon after hatch. Artificially increasing microbial diversity early in life may aid in the promotion of positive microorganism-microorganism and microorganism-host interactions that promote immune development and resistance to important diseases, including NE. Strategies to cost-effectively apply a microbiota that confers a health benefit (e.g. administration on eggs), and validation of efficacy in production settings is required. It is anticipated that the finds from the current study will facilitate studies to address these topics.”

Despite the advantages of including more birds in the study, we feel that our findings are statistically valid, and that the scientific community should see our work with the goal of potentially stimulating further research on the use of microbiota transplants in day-old chicks against necrotic enteritis, and potentially other diseases.

  1. Zaytsoff, S.J.M.; Boras, V.F.; Uwiera, R.R.E.; Inglis, G.D. A stress-induced model of acute necrotic enteritis in broiler chickens using dietary corticosterone administration. Poultr Sci 2022, 101, 101726.

Reviewer comment. Which stat was used for lesion scores analysis? The most appropriate is Fisher's exact test (2x2 contingency table). I am not sure how you can see a significance with just 6 biological reps.

Author response. Kruskal-Wallis test, which along with Fisher’s exact test, is a non-parametric test was used to analyze histopathologic data. This is specified in the Materials and Methods section.

Reviewer comment. Authors also need to mention if this model simulates a 'subclinical' or a 'clinical' NE. Based on the average lesion score, it looks mild but again it's very difficult since only 1 bird had 5 score. Please mention the source of CP1 and looks like it has a 'low' level of virulence in vivo.

Author response. CP1 is a highly virulent strain of C. perfringens, provided by John Prescott and his team [2]. It is an acute model, and we cite the manuscript in which the model of necrotic enteritis used in the current study is described [1]; the title of this article is “A stress-induced model of acute necrotic enteritis in broiler chickens using dietary corticosterone administration”.

  1. Zhou, H.; Lepp, D.; Pei, Y.; Liu, M.; Yin, X.; Ma, R.; Prescott, J.F.; Gong, J. Influence of pcp1netb ancillary genes on the virulence of Clostridium perfringens poultry necrotic enteritis strain cp1. Gut Pathog 2017, 9, 6.

Reviewer comment. Authors need to mention how many birds per treatment used? Not sure why I wasn't able to find it in the M&M design.

Author response. This was specified in the “Experimental Design” subsection of the Materials and Methods section.

Reviewer comment. 2. Page 337: NE amelioration was observed only in jejunum? We always find lesions in duodenum and/or jejunum under experimental NE setting. Page 358-359: The premise that MT can displace pathogenic C. perfringens MAY be a little flawed since certain CP strains carry signature virulence genes that are absent in commensal CP and such virulent strains don't usually reside as opportunistic CP but gain entry upon intestinal predisposition.

Author response. The CP1 strain of C. perfringens used is a virulent strain that produces the NetB toxin. In the last paragraph of the introduction we indicated the following, “To facilitate the evaluation of the impacts of treatments on acute disease, corticosterone was provided in the diet to mediate a stress response in birds and promote the onset of clinical NE upon challenge with C. perfringens [20].” Citation 20 is reference to our manuscript describing the model used (i.e. reference #1 herein).

Reviewer 3 Report

This ms. submission by Zaytsoff et al describes the use of microbiota transplants in day old chicks to ameliorate the effects of C.perfringens induced NE. The ms. is well written and the results are quite extensive, with many aspects of the microbiota, metabolomics and host gene expression levels included. The results are statistically significant, although barely so in several experiments and the overall conclusions drawn from the study are of significance to the field. The use of bioreactors to propagate the MTs seems justified, since this is probably what would be used on a commercial scale anyway. Overall, a very solid paper with significant findings.

Author Response

Comments and Suggestions for Authors

Reviewer comment. This ms. submission by Zaytsoff et al describes the use of microbiota transplants in day old chicks to ameliorate the effects of C.perfringens induced NE. The ms. is well written and the results are quite extensive, with many aspects of the microbiota, metabolomics and host gene expression levels included. The results are statistically significant, although barely so in several experiments and the overall conclusions drawn from the study are of significance to the field. The use of bioreactors to propagate the MTs seems justified, since this is probably what would be used on a commercial scale anyway. Overall, a very solid paper with significant findings.

Author response. We thank the reviewer for the positive comments.